# Bioinspired Swimming Robots with 3D Biomimetic Shark Denticle Structures for Controlled Marangoni Propulsion

**DOI:** 10.3390/biomimetics10080479

**Published:** 2025-07-22

**Authors:** Kang Yang, Chengming Wang, Lei Jiang, Ruochen Fang, Zhichao Dong

**Affiliations:** 1CAS Key Laboratory of Bio-Inspired Materials and Interfacial Science, Technical Institute of Physics and Chemistry, Chinese Academy of Sciences, Beijing 100190, China; yangkang20@mails.ucas.ac.cn (K.Y.); wangchengming23@mails.ucas.ac.cn (C.W.); jianglei@iccas.ac.cn (L.J.); 2School of Future Technology, University of Chinese Academy of Sciences, Beijing 100049, China; 3International Institute for Interdisciplinary and Frontiers, Beihang University, Beijing 100191, China

**Keywords:** shark denticles, Marangoni propulsion, superhydrophobic, drag reduction

## Abstract

Shark skin exhibits a well-defined multilayered architecture, consisting of three-dimensional denticles and an underlying dermal layer, which contributes to its passive drag reduction. However, the active drag reduction mechanisms of this interface remain largely unexplored. In this study, the Marangoni effect potentially arising from the active secretion of mucus on shark skin is investigated. A 3D-printed swimming robot with a porous substrate and a biomimetic shark denticle structure is developed. By introducing surfactants into the porous substrate and adjusting denticle arrangements, on-demand propulsion and controlled swimming trajectories are achieved. A superhydrophobic surface is fabricated on the swimming robot, which reduces water resistance and enhances propulsion. Moreover, denticles with a 30° attack angle demonstrate optimal propulsion performance in both Marangoni-driven hydrodynamics and aerodynamics. This study suggests that the secretion of mucus on shark skin may facilitate active drag reduction via the Marangoni effect, offering novel insights into the biomimetic structural design of autonomous swimming robots.

## 1. Introduction

Over the past two decades, significant research efforts have been directed toward developing untethered miniature robots, ranging from micrometers to centimeters in size, capable of autonomous movement [1,2], controllable manipulation [3,4], and remote communication [5]. These small-scale robots hold immense promise for applications such as environmental monitoring [6], sensing [7,8], drug delivery [9,10], environmental toxin remediation [11,12], and energy generation [13,14]. Inspired by the intricate locomotion systems of insects, engineers have designed innovative robots that replicate these natural mechanisms. For instance, rove beetles such as *Microvelia* and *Stenus* can glide across water surfaces by secreting chemicals from their rear, creating a surface tension gradient wherein a liquid with lower surface tension is drawn by the surrounding higher surface tension liquid, resulting in rapid Marangoni propulsion [15]. This bioinspired principle has driven the development of self-propelled swimming robots, powered by diverse actuation methods, such as magnetic [16,17,18], thermal [19,20,21], electric [22,23,24], and concentration fields [25,26,27,28]. However, existing Marangoni-propelled swimmers consisting of single materials fall short of achieving multiple functionalities, and structural design has emerged as a critical factor affecting the performance of Marangoni-propelled swimmers, particularly in manipulating surface tension dynamics. While current efforts in structural design have been directed toward the whole swimmer through combining different components, there remains a lack of in-depth investigation into the individual structural elements and their micro-nanoscale features, which are essential for optimizing propulsion and functionality [29].

In nature, organisms secrete a variety of chemical substances to perform diverse functions. Fish, for instance, release mucus on their surfaces to reduce drag, and sharks similarly produce mucus, though current research suggests that the amount secreted is minimal and may not significantly contribute to drag reduction [30,31,32]. Studies on shark drag reduction have predominantly focused on the passive effects of their denticles [31,33,34,35,36,37,38,39,40,41,42,43,44,45], with minimal attention devoted to active mechanisms involving mucus (see Appendix A for a comparison of drag reduction approaches across different studies). However, some evidence indicates that the mucus secreted by fish exhibits a low surface tension, typically ranging from 20 to 30 mN/m [46,47,48,49], which is comparable to that of certain chemical fuels, such as ethanol with a surface tension of 22.39 mN/m. This similarity suggests that sharks may utilize an active drag-reduction strategy through the secretion of a specific quantity of mucus from their skin and gills. The low surface tension of this mucus could induce Marangoni propulsion, facilitating active drag reduction. When combined with the hydrodynamic properties of the denticle micro-structures, this mechanism could potentially yield unexpectedly enhanced performance, warranting further investigation into such bioinspired active drag-reduction approaches.

Here, to investigate the potential Marangoni effect induced by mucus on shark skin, we utilized simple and efficient 3D printing technology to fabricate a swimming robot featuring a porous substrate topped by a denticle array structure. By introducing a surfactant through the pores between the denticles, Marangoni propulsion was achieved as the surfactant was released. Through careful adjustment of the denticle arrangement and incorporation of corresponding fluid diversion channels, untethered propulsion and controlled directionality of motion were achieved. Additionally, water resistance was significantly reduced by integrating superhydrophobic denticles and fences along the robot’s sides, further enhancing propulsion efficiency. Denticles designed with a 30° attack angle exhibited optimal propulsion performance in both Marangoni-propelled hydrodynamics and aerodynamics. This study introduces a novel concept of biomimetic design for autonomous swimming robots, highlighting the possible role of shark mucus in active drag reduction through the Marangoni effect.

## 2. Materials and Methods

### 2.1. Fabrication of 3D-Printed Self-Propelled Swimmer

Freshly deceased specimens of a shortfin mako shark (*Isurus oxyrinchus*) and a Pacific spadenose shark (*Scoliodon macrorhynchos*), were sourced from fishermen in Shandong Province, China. A small piece of skin, measuring 2 × 3 mm^2^, from the flank region of the specimen was used for micro-CT scanning (SKYSCAN 1272, Bruker, Billerica, MA, USA). A single representative denticle was selected from the scanning sample, and a 3D digital mesh model was constructed using Mimics Research 21.0 (Materialise Inc., Leuven, Belgium).

The reconstructed denticle model was duplicated and arranged in a staggered and overlapped pattern and linear pattern, similar to real shark denticles, where the riblet tips of the anterior denticle fall into or interlock with the valleys of the posterior denticle, onto a substrate using 3ds Max (Autodesk, Inc., San Francisco, CA, USA). The real denticle height (1.1 mm) was reduced due to its embedded height in the substrate. The denticles and porous substrate (3 cm × 3 cm × 2 mm) with diversion walls were printed using a commercial digital light processing printer (Mini 8K, Phrozen Tech., Taiwan, China) with an X-Y axes resolution of 22 μm and *Z*-axis resolution of 10 μm, using a methyl methacrylate-SiO_2_ mixed resin. The printing layer height was set at 20 μm, with an exposure time of 2.4 s. After 3D printing, the samples were immersed in ethanol for 10 min and subjected to ultrasonic cleaning to remove uncured resin. To enhance the mechanical properties, a post-curing process was performed with LEDs emitting 405 nm light for 3 min at room temperature. Due to limitations in printing resolution, all 3D-printed denticles were scaled up to ten times the actual size of real denticles. To obtain the swimmer with superhydrophobic fences and denticles, the surfaces were carefully brushed with a superhydrophobic solution consisting of 1.0 g of superhydrophobic nanosilica particles (Aerosil R202, average particle size 14 nm, Evonik Degussa Co., Essen, Germany) in 100 mL of n-hexane using a hairbrush and dried under ambient conditions for 10 min to ensure the solvent was completely volatilized.

### 2.2. Characterization

SEM images were obtained using a field-emission scanning electron microscope at 10 kV (SU8010, Hitachi, Ibaraki Prefecture, Japan). The sharkskin samples were first rinsed in an ultrasonic cleaner with water for 30 s and then cleaned with deionized water three times to remove mucus, dirt, and blood on the surface. Tissue samples were then fixed with 2.5% glutaraldehyde overnight at 4 °C. After fixation, the samples were rinsed with phosphate buffer solution and then rinsed with deionized water three to five times. Dehydration of the tissue was performed by the application of a consecutive series of increased concentrations of ethanol (30%, 50%, 75%, 90%, and 100%) for 15 min each in order to avoid shrinkage and deformation. Finally, the samples were air-dried under a vacuum before being sputter-coated with gold. The water contact angles were measured using a video-based contact angle measuring device (OCA 25, Data-physics, Filderstadt, Germany) with droplets of 2.0 μL. The surface morphology of the 3D-printed Pacific spadenose shark denticles was characterized by OLYMPUS OLS-4500 (Olympus Corporation, Tokyo, Japan). An atomic force microscopy (AFM) image was obtained using an atomic force microscope (Dimension FastscanBio, Bruker, Billerica, MA, USA).

### 2.3. Propulsion Performance Analysis

For hydrodynamic locomotion, the swimmer was placed in a water tank with an area of 50 × 50 cm^2^. Locomotion of the swimmer was recorded with a Nikon D750 camera (Nikon, Tokyo, Japan) from the top view. Tracker software (7.6.21) was used to obtain coordinates of the mass center, symmetric axis, and trajectory curvature of the swimmers. For aerodynamic locomotion, the swimmer was connected to a wall-mounted stress-strain sensor (with an accuracy of 1 mN and a measuring range of 0–10 N) to measure the total propulsion force generated during actuation. A pressure-controlled air compressor was connected to a hole on the top of the swimmer to provide a steady airflow input. The propulsion forces were recorded under various airflow pressures to evaluate the aerodynamic thrust performance and efficiency of the swimmer design.

## 3. Results and Discussion

### 3.1. Concept and Design Approach

By integrating the shark denticle structure and cutaneous mucus secretion, we propose here a hybrid swimmer design consisting of a porous substrate infused with surfactant for propulsion and a denticle array equipped with fluid diversion channels for directional control. Figure 1a displays a schematic diagram of a shortfin mako shark (*Isurus oxyrinchus*), with its corresponding denticle structure (Figure 1b). A side view reveals the denticles and skin, where the skin comprises a thin epidermis overlaying a thicker dermis, the latter divided into a superficial stratum laxum and a deeper stratum compactum. Each denticle consists of a crown, neck, and base, with a distinct cavity region between the skin and the crown. The epidermis, composed of only a few cell layers, is situated between the denticles and contains mucus-producing goblet cells, which are less abundant in the shortfin mako compared to other elasmobranch fishes (Figure 1c). Although the shortfin mako sharks are believed to produce less mucus, the role of this mucus in drag reduction remains unclear. Additionally, studies indicate that denticles can tilt to angles exceeding 50°, providing a passive mechanism to manage flow separation and prevent reverse flow (Figure 1d). This dynamic adjustment of denticle angle during swimming likely influences the distribution of mucus on the denticles, as well as the diffusion rate of mucus into the surrounding water, thereby affecting the overall Marangoni propulsion process. In the following sections, we incorporate these biological features into the design of self-propelled swimming robots to investigate their effects on propulsion performance.

### 3.2. Design and Hydrodynamics Locomotion of Self-Propelled Swimmers

To fabricate a self-propelled swimmer (3 × 3 cm^2^) with a denticle array on a porous substrate, we utilized resin-based 3D printing technology to construct a hybrid bilayer structure. The outermost denticle layer was printed using a 3D model reconstructed from micro-computed tomography (micro-CT). Additional details regarding the 3D reconstruction of a single denticle are provided in our previous study [50]. Each denticle features two through-holes beneath it, collectively forming an array aligned with the denticle arrangement (Appendix A). Figure 2a illustrates the propulsion mechanism of the self-propelled Marangoni swimmer. The 3D-printed swimmer was positioned stationary on the water surface, and a precise volume of surfactant was introduced into the through-holes using a pipette. The surfactant flowed through the holes into the gaps between the denticles, ultimately contacting the water to induce Marangoni convection.

To enhance propulsion and control, superhydrophobic fences were incorporated along the sides and head of the swimmer (Figure 2b and Appendix A). The hydrophilic denticles were also rendered superhydrophobic (Appendix A). Initial tests were conducted on staggered and overlapped denticle arrays. When 2.5 µL of hexafluoroisopropanol (HFIP) was introduced into the central through-holes on the bottom of the swimmer, the swimmer achieved self-propulsion, but its trajectory was erratic, and its speed fluctuated (Figure 2c and Appendix A). This behavior resulted from the staggered and overlapped denticle arrangement, combined with the surface tension of the surfactant, causing non-uniform diffusion of the surfactant into the cavity region beneath the denticles. This resulted in heterogeneous surfactant concentrations underneath each denticle, generating non-uniform Marangoni thrust. However, when surfactant was introduced into all through-holes in the left half of the staggered and overlapped denticle array, the swimmer exhibited certain clockwise rotational motion with progressively stabilizing velocity (Figure 2d and Appendix A). Similarly, introducing surfactant into the right half of the through-holes enabled counterclockwise rotational motion (Appendix A).

To achieve controlled and predictable motion of the swimmer, we optimized the denticle arrangement. The staggered and overlapped denticle configuration was replaced with a linear arrangement, incorporating diversion walls between every two columns of denticles (Appendix A). This design mitigated non-uniform surfactant diffusion, ensuring that the surfactant remained concentrated within a single column of denticles. The swimmer was divided into 16 denticle columns, and introducing surfactant into the 8th and 9th columns enabled linear forward motion, with the swimmer accelerating initially before maintaining a constant velocity until it reached the edge of the water tank (Figure 3a and Appendix A). Similarly, adding surfactant to a single column on the left side induced consistent clockwise rotational motion with a progressively decreasing oscillatory velocity (Figure 3b and Appendix A). Likewise, introducing surfactant to a right-side column resulted in consistent counterclockwise rotational motion (Appendix A).

To enhance the propulsion performance of the linearly arranged denticle swimmer, we rendered the denticles and fences superhydrophobic and compared its performance to that of hydrophilic ones. As the surfactant volume increased, the swimmer’s velocity also increased. The superhydrophobic-treated swimmer achieved a higher velocity of 8.63 cm s^−1^ compared to the hydrophilic counterpart (Figure 3c) and sustained propulsion for more than twice as long (Appendix A). We also assessed the propulsion effects of various surfactants, including hexafluoroisopropanol (HFIP), isopropanol (IPA), n-propanol, ethanol (EtOH), and dimethyl sulfoxide (DMSO). As shown in Figure 3d, HFIP, with the lowest surface tension of 14.53 mN/m, yielded the best propulsion performance, while DMSO, with a surface tension of 43.60 mN/m, demonstrated the lowest performance. Appendix A presents the relationship between the surface tension of different surfactants and propulsion velocity, from which an exception is noted with N-propanol and ethanol. Despite ethanol having a lower surface tension (22.39 mN/m) than N-propanol (23.78 mN/m), its propulsion velocity is slower. This anomaly may be attributed to ethanol’s faster evaporation rate, which results in a shorter maintenance of the surface tension gradient, leading to rapid dilution of the propulsion region and reduced propulsion force. Additionally, factors such as surfactant wettability, surface morphology, and wetting behavior may also influence propulsion performance. Nevertheless, this indicates that surfactants with lower surface tension result in better propulsion performance. Additionally, we investigated the effect of varying denticle attack angles on propulsion, finding that an angle of 30° resulted in the highest velocity of 10.05 cm s^−1^, followed by angles of 14° and 45°. Superhydrophobic-treated swimmers consistently outperformed their hydrophilic counterparts across these angles (Figure 3e). Finally, we evaluated both superhydrophobic and hydrophilic swimmers with the five chemical fuels at various attack angles, confirming that the 30° attack angle consistently achieved the highest propulsion velocity regardless of the chemical fuel used (Figure 3f,g). The efficiency of the swimmer is calculated as the kinetic energy of the swimmer (which depends on speed and the total mass of the swimmer) per unit mass of fuel. For the highest-performing swimmer using HFIP as fuel, 2.5 μL of fuel yielded an efficiency of 1879 μJ g^−1^. The reason for this high efficiency may be that the fuel forms vortices in the cavity region underneath the denticles, enhancing the exchange between fluids. Overall, the fabricated self-propelled swimmer demonstrated on-demand propulsion and controlled swimming trajectories, with optimal propulsion performance at a 30° attack angle. These findings suggest that mucus with a surface tension comparable to chemical fuels, such as ethanol, may generate a Marangoni effect, thereby enabling active drag reduction and propulsion.

### 3.3. Design and Aerodynamics Locomotion of the Swimmers

To evaluate the propulsion performance of the swimmer under aerodynamic conditions, thrust generation was measured. An air inlet was incorporated at the bottom center of the previously developed Marangoni self-propelled swimmer, which connected to an air compressor. Compressed air entered an internal chamber within the swimmer, flowed through the through-holes beneath the denticles, and was discharged between them. A stress–strain sensor was mounted at the anterior end of the swimmer to measure thrust magnitude (Figure 4a). By varying air pressure, we observed that thrust increased with elevated pressure, with the maximum thrust achieved at a denticle attack angle of 30°, followed by angles of 14° and 45° (Figure 4b). These results, consistent across both hydrodynamic and aerodynamic conditions, confirm that a 30° attack angle yields optimal propulsion performance. And the greater the external propulsion force (air pressure or fuel), the more significant the propulsion effect. Additionally, we investigated the propulsion performance of another shark species, the Pacific spadenose shark (*Scoliodon macrorhynchos*), and found it to be inferior to that of the shortfin mako shark (*Isurus oxyrinchus*) (Figure 4c). This difference may be attributed to the larger denticle size of the Pacific spadenose shark relative to the shortfin mako (Appendix A), resulting in a reduced pore density per unit area(1)$$\varepsilon =N_{pores}/A$$
where A is the total surface area of the swimmer and $N_{pores}$ is the number of pores. The pore density for the shortfin mako shark was 126 pores cm^−2^, while that for the Pacific spadenose shark was 79.3 pores cm^−2^, suggesting that higher pore density facilitates greater interaction between air and denticles, thereby generating greater thrust.

Finally, further investigation into the propulsion performance of the Pacific spadenose shark revealed a distinct hexagonal pit structure on its denticles, with pit heights ranging from 2–3 μm (Appendix A). The effect of these hexagonal pits on drag reduction or propulsion remains uninvestigated. To investigate this, we fabricated biomimetic denticles with hexagonal structures using 3D printing (Appendix A) and evaluated their propulsion performance. Compared to denticles without hexagonal structures, those with hexagonal pits exhibited no enhancement in propulsion performance under aerodynamic conditions (Figure 4d) but demonstrated improved Marangoni-driven hydrodynamic propulsion (Figure 4e). Subsequently, we tested the drag of denticles with and without hexagonal pit structures in a circulating water tunnel, finding that denticles with hexagonal pits exhibited superior drag reduction (Appendix A). This result may be attributed to reduced vortex formation [50] in the cavity regions of denticles with hexagonal pits under hydrodynamic conditions. Specifically, when surfactant enters the cavity regions of these denticles, hexagonal pits may retain and direct surfactant flow into the water, facilitating controlled release, suppressing reverse flow, and thereby minimizing vortex formation. In contrast, under aerodynamic conditions, the thrust generated by denticles with and without hexagonal pits was comparable, likely because gases are expelled as jets from the cavity regions under high pressure in both cases. Future studies will focus on the detailed fluidic characterization of surfactant dynamics in the cavity regions and the effects of hexagonal pit structures on Marangoni propulsion and drag reduction. Overall, both hydrodynamic and aerodynamic tests consistently demonstrated that a 30° attack angle provides optimal propulsion performance, with greater external propulsion (fuel or air pressure) enhancing the performance.

## 4. Conclusions

In summary, we 3D-printed and analyzed the Marangoni propulsion performance of the self-propelled swimmer featuring a porous substrate topped by a denticle array structure, inspired by the denticle structure and mucus secretion of sharks. By introducing surfactants through the pores beneath the denticles, Marangoni propulsion was successfully generated, enabling untethered motion. Optimized adjustments to the denticle arrangement, including the transition from staggered and overlapped to linear configurations, coupled with the incorporation of fluid diversion channels, enabled on-demand propulsion and controllable manipulation over swimming trajectories. The integration of superhydrophobic denticles and fences along the swimmer’s sides significantly reduced water resistance, enhancing propulsion efficiency, with superhydrophobic-treated swimmers achieving a maximum velocity of 10.05 cm s^−1^ and sustaining propulsion for over twice as long as their hydrophilic counterparts. Denticles with a 30° attack angle demonstrated optimal propulsion performance in both Marangoni-driven hydrodynamic and aerodynamic conditions. Comparative analysis of shark species revealed that the shortfin mako shark exhibited superior propulsion performance compared to the Pacific spadenose shark under aerodynamic conditions. Additionally, the hexagonal pit structures on Pacific spadenose shark denticles, with pit heights of 2–3 μm, exhibited notable enhancement in Marangoni-driven hydrodynamic propulsion performance. This study presents a novel biomimetic design for autonomous swimming robots, highlighting the potential role of shark mucus in active drag reduction via the Marangoni effect and providing a foundation for future exploration of micro-nanoscale structural influences on propulsion performance.

## Figures and Tables

**Figure 1 biomimetics-10-00479-f001:**
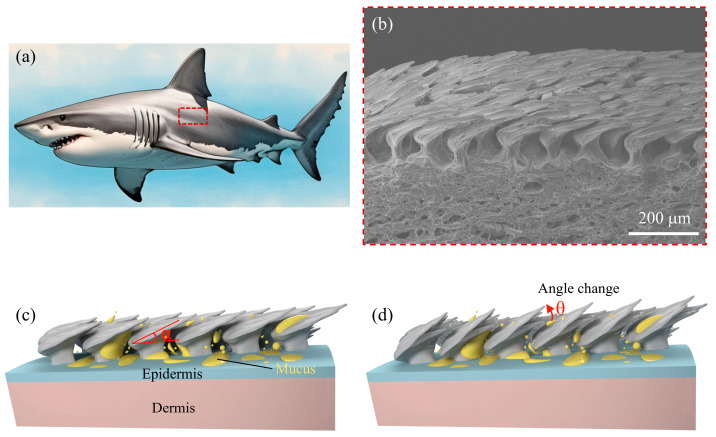
Shark denticle characterization and propulsion design concept. (**a**) Schematic of the shortfin mako shark (*Isurus oxyrinchus*). (**b**) Cross-sectional scanning electron microscopy (SEM) image of shark denticles. (**c**) Schematic illustration of the hierarchical structure of shark skin, comprising denticles, epidermis, and dermis, along with the distribution of mucus. Denticles are initially oriented at an attack angle α. (**d**) Variation in attack angle θ in response to changing flow conditions.

**Figure 2 biomimetics-10-00479-f002:**
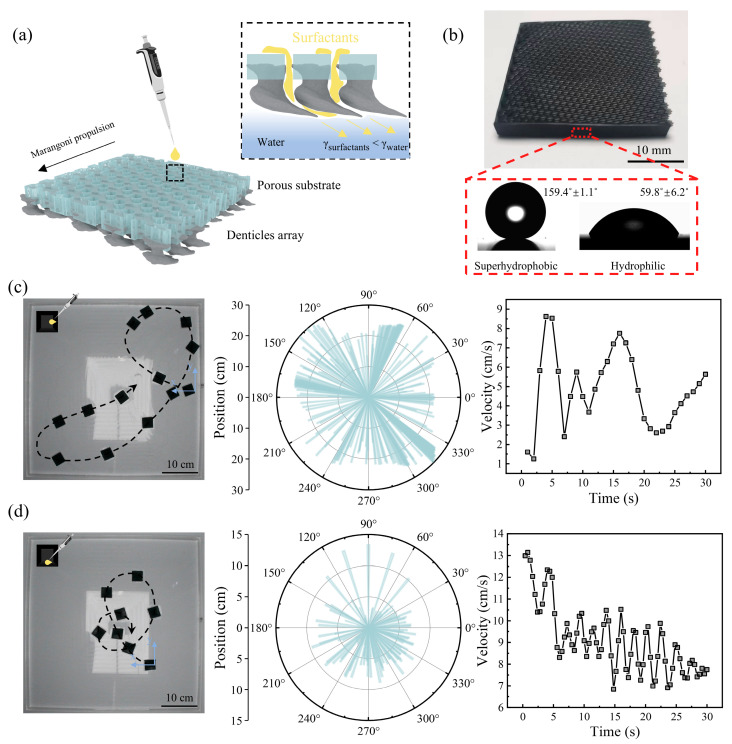
(**a**) Self-propelled Marangoni-driven swimmer. (**b**) Photograph of the 3D printed self-propelled Marangoni swimmer with staggered and overlapped denticle arrays, featuring superhydrophobic or hydrophilic fences along sides of the swimmer. (**c**) Self-propelled Marangoni swimmer with staggered and overlapped denticle arrays exhibits unstable trajectories when HFIP is introduced into the central through-holes on the bottom of the swimmer. (**d**) Self-propelled Marangoni swimmer with staggered and overlapped denticle arrays exhibits certain clockwise rotational motion when HFIP is introduced into all through-holes in the left half of the swimmer.

**Figure 3 biomimetics-10-00479-f003:**
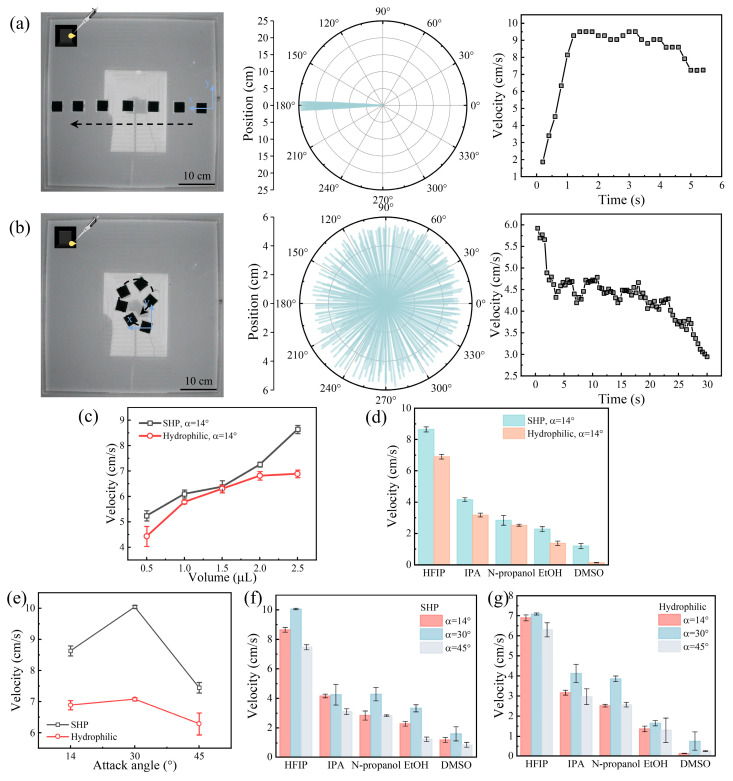
(**a**) Self-propelled Marangoni swimmer with a linear denticle arrangement exhibiting linear forward motion when HFIP is introduced into the central holes at the bottom of the swimmer. (**b**) Self-propelled Marangoni swimmer with a linear denticle arrangement exhibiting continuous and regular clockwise rotation when HFIP is introduced into the left column of the swimmer. (**c**) Propulsion velocity of superhydrophobic and hydrophilic swimmers with a linear denticle arrangement at varying HFIP volumes. (**d**) Propulsion velocity of superhydrophobic and hydrophilic swimmers with a linear denticle arrangement using various surfactants. (**e**) Propulsion velocity of superhydrophobic and hydrophilic swimmers with a linear denticle arrangement at varying attack angles. (**f**) Relationship between the propulsion velocity of the superhydrophobic swimmer and surfactant type at varying attack angles. (**g**) Relationship between the propulsion velocity of the hydrophilic swimmer and surfactant type at varying attack angles.

**Figure 4 biomimetics-10-00479-f004:**
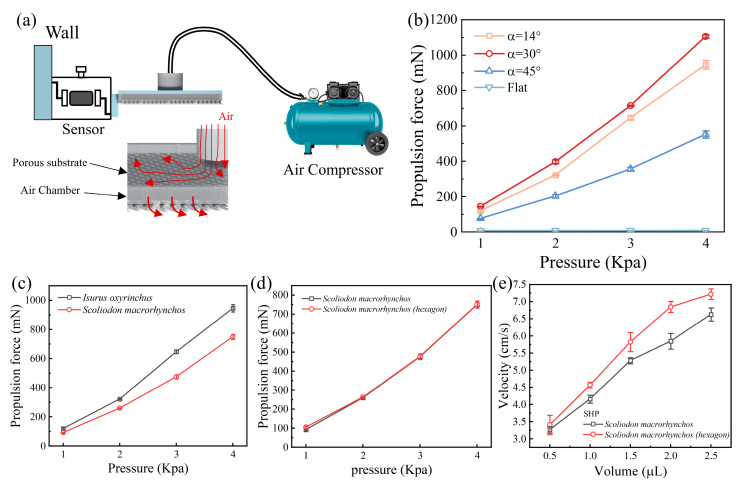
(**a**) Schematic illustration of the aerodynamic propulsion of the swimmer. (**b**) Relationship between propulsion force and air pressure at varying attack angles. (**c**) Relationship between propulsion force and air pressure for denticle models of different shark species. (**d**) Relationship between propulsion force and air pressure for *Scoliodon macrorhynchos* denticles (with and without hexagonal structures). (**e**) Relationship between propulsion velocity and HFIP volume for *Scoliodon macrorhynchos* denticles (with and without hexagonal structures).

## Data Availability

Data are contained within the article and Appendix A.

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
