# Peer review of "Bioinspired Swimming Robots with 3D Biomimetic Shark Denticle Structures for Controlled Marangoni Propulsion"

_biomimetics, 2025, doi:10.3390/biomimetics10080479_

Round 1
Reviewer 1 Report
Comments and Suggestions for Authors
In the manuscript titled “Bioinspired Swimming Robots with 3D Biomimetic Shark Denticle Structures for Controlled Marangoni Propulsion”, the authors present a 3D-printed swimming robot featuring a porous substrate combined with shark-inspired denticle structures to enable on-demand Marangoni propulsion. By tuning the surfactant release and the tilt angles of the denticles, the robot achieves a peak velocity of 10 cm/s. The design is both innovative and timely, supported by thorough experimental analysis. I recommend the manuscript for publication after minor revisions.
Comments:
- The key geometrical parameters of the Marangoni-driven swimmer, such as the height, length of the 3D printed denticle should be provided in the manuscript.
- For the linear swimmer, how was the spacing between denticles selected? Additionally, how does this spacing influence the swimmer's propulsion velocity?
- In figure 3, please clarify the durability and longevity of the Marangoni-driven swimmer.
- The relationship between the Marangoni-driven swimmer and aerodynamically propelled swimmers should be further strengthened.
Reviewer 2 Report
Comments and Suggestions for Authors
This study presents
a biomimetic design of autonomous swimming robots, highlighting the potential role of
shark slime that is actively reducing drag due to the Marangoni effect. The topic of the work is of interest, however, its degree of novelty remains unclear, since there are no graphic materials that would illustrate a comparison with previously known results, for example, with works cited in the list of references numbered 31-37. In addition, the article does not provide the necessary description of both the Marangoni effect itself and the mechanism for reducing drag due to this effect, for example, depending on the amount of surface tension of the surfactant and its amount. In the present form, I consider the publication of this work to be inappropriate, since it requires substantial revision, taking into account the comments noted above.
Round 2
Reviewer 2 Report
Comments and Suggestions for Authors
The authors do not take into account the comments at all, limiting themselves to only minor clarifications that do not clarify the issues raised in the previous review. In the present form, I consider it inappropriate to publish this material.
Round 3
Reviewer 2 Report
Comments and Suggestions for Authors
Now it is possible to publish material in the present form.